# Making the Best Use of Available Weapons for the Inevitable Rivalry-Resistance to EGFR-TKIs

**DOI:** 10.3390/biomedicines11041141

**Published:** 2023-04-10

**Authors:** Dongyu Li, Jingnan Wang, Chengming Liu, Yuejun Luo, Haiyan Xu, Yan Wang, Nan Sun, Jie He

**Affiliations:** 1Department of Thoracic Surgery, National Cancer Center/National Clinical Research Center for Cancer/Cancer Hospital, Chinese Academy of Medical Sciences and Peking Union Medical College, Beijing 100021, China; 2State Key Laboratory of Molecular Oncology, National Cancer Center/National Clinical Research Center for Cancer/Cancer Hospital, Chinese Academy of Medical Sciences and Peking Union Medical College, Beijing 100021, China; 34 + 4 Medical Doctor Program, Chinese Academy of Medical Sciences & Peking Union Medical College, Beijing 100021, China; 4Department of Radiation Oncology, National Cancer Center/National Clinical Research Center for Cancer/Cancer Hospital, Chinese Academy of Medical Sciences and Peking Union Medical College, Beijing 100021, China; 5Department of Comprehensive Oncology, National Cancer Center/National Clinical Research Center for Cancer/Cancer Hospital, Chinese Academy of Medical Sciences and Peking Union Medical College, Beijing 100021, China; 6Department of Medical Oncology, National Cancer Center/National Clinical Research Center for Cancer/Cancer Hospital, Chinese Academy of Medical Sciences and Peking Union Medical College, Beijing 100021, China

**Keywords:** EGFR-TKIs, NSCLC, lung cancer, sequential therapy, resistant mechanisms

## Abstract

The emergence of epidermal growth factor receptor-tyrosine kinase inhibitors (EGFR-TKIs) revolutionized the treatment of advanced-stage non-small cell lung cancer (NSCLC). Detected in more than 50% of late-stage lung adenocarcinoma in Asian patients, the EGFR mutation was regarded as a golden mutation for Asians. However, resistance to TKIs seems inevitable and severely hinders patients from getting further benefits from treatment. Even though resistance caused by EGFR T790M could be effectively managed by third-generation EGFR-TKIs currently, resistance to third-generation EGFR-TKIs is still a troublesome issue faced by both clinicians and patients. Various efforts have been made to maximize the benefits of patients from EGFR-TKIs therapy. Thus, new requirements and challenges have been posed to clinicians of this era. In this review, we summarized the clinical evidence on the efficacy of third-generation EGFR-TKIs in patients with EGFR-mutated NSCLC. Then, we discussed advancements in sequential treatment aiming to delay the onset of resistance. Moreover, the resistance mechanisms and features were depicted to help us better understand our enemies. Lastly, we put forward future strategies, including recent approaches involving the utilization of antibody drug conjugates against resistance and research directions about shaping the evolution of NSCLC as a core idea in the management of NSCLC.

## 1. Introduction

Non-small cell lung cancer (NSCLC) is one of the most aggressive cancer types and the leading cause of cancer related death in China as well as world-wide [1,2]. Most patients in China are already at unresectable late stage at diagnosis and the efficacy of chemotherapy and radiotherapy were unsatisfying. Fortunately, the advancement of EGFR-TKIs dramatically revolutionized treatment strategies for a great proportion of NSCLC patients carrying EGFR mutations (Figure 1), which is the most common oncogenic mutations among Asians with lung adenocarcinoma (LUAD) [3,4,5]. Among the patients carrying EGFR mutations, roughly 85% patients carry the classic mutations, 19deletions (19dels) and 21exon L858R point mutation. The rest 15% uncommon mutations include exon 19 insertion, exon 18 G719X, and exon20 S768I, which would respond relatively poorly to first, second, and third generation TKIs [6,7]. Based on the results from previous clinical trials, the remission rate for these subgroups was roughly 0–11% [8,9]. Multiple ongoing trials are recruiting patients to explore the efficacy of novel drugs such as mobocertinib and JMT-101 for uncommon mutation, we believe these trials will provide valuable evidence for the management of these patients in the future [10]. Exon 20 insertion, which accounted for up to 3.5% of all EGFR mutated NSCLC patients, was once regarded as an uncommon mutation that could not be treated by EGFR-TKIs [11]. However, based on the inspiring results from CHRYSALIS study, FDA approved Amivantamab and mobocertinib for the management of patients with this mutation, which is a major breakthrough for Exon 20 insertions [12,13]. Details about Amivantamab and mobocertinib would be discussed later in this review. An overview of different generations of EGFR-TKIs available worldwide is displayed in Table 1.

Multiple prospective clinical trials and real-world studies have confirmed the superiorities of EGFR-TKIs regarding objective response rate (ORR), progression free survival (PFS), overall survival (OS), and quality of life (QoL) when treating late-stage NSCLC patients with EGFR-sensitive mutations compared to conventional platinum-based Chemotherapy [14,15,16,17]. As a result, screening for possible EGFR mutations has been a standard of care for patients with advanced lung adenocarcinoma and EGFR-TKIs have been approved as first-line therapy for patients carrying EGFR-sensitive mutations by several authoritative oncology associations such as ASCO, ESMO, and NCCN [18,19]. However, resistance to first- and second-generation EGFR-TKIs will inevitably develop after 9–12 months in median, and patients will no longer benefit from TKI therapy [20,21]. Even though T790M mutation, the most common acquired resistant mechanism after first- or second-generation EGFR-TKIs, could now be effectively managed using the third generation EGFR-TKIs such as osimertinib, furmonertinib and aumolertinib [17]. Resistance to third generation EGFR-TKIs is still a major obstacle which hinders patients to further benefit from TKI therapy.

As resistance to EGFR-TKIs seems inevitable, attempting different drug combinations in sequential treatment aiming to extend treatment to resistance duration as well as strategies to overcome resistance have become major focus of research in this area. This review summarized recent clinical advancements regarding different drug combinations in sequential treatment. Moreover, we further introduced common mechanisms leading to resistance as well as progresses made to overcome resistance, aiming to provide support for clinicians and insights about future research in this field. In order to truly cure NSCLC, we could not only try to chase the evolution of lung cancer but should manage to shape their evolution.

## 2. Conversant with Our Weapons: Clinical Data of Third-Generation EGFR-TKIs

Third generation EGFR-TKIs, characterized by osimertinib, furmonertinib, and aumolertinib were designed specifically to combat against T790M resistance, which is the most common type of resistance after first- or second-generation EGFR-TKIs treatment [22]. In silica study indicates affinity between osimertinib and EGFR T790M is nearly 200-fold higher than wild type, confirming its high selectivity [23]. According to the results of AURA1 and AURA3, utilization of osimertinib for patients developed T790M acquired resistance after first or second generation TKIs treatment could effectively extends PFS and OS comparing to platinum-based chemotherapy. Moreover, cytotoxicity and rate of adverse events of osimertinib were significantly lower than platinum-based chemotherapy [17,24,25]. More importantly, osimertinib also brought hope for the treatment of advanced NSCLC with brain metastasis. Based on chemical structure of osimertinib, it has higher permeability for blood brain barrier and greater retention within the compared to first and second generations EGFR-TKIs, therefore, it could effectively control brain metastasis with EGFR-sensitive mutation [17]. This hypothesis was further confirmed by later BLOOM Phase 1 trials [26]. According to third party independent review, meningeal ORR was 62% and Duration of Response (DoR) 15.2 months. Meanwhile, AURA3 and OCEAN study also confirmed that osimertinib could effectively reduce the rate of brain metastasis, control the growth of cranial metastasis, and extends PFS and OS [27,28]. Based on these inspiring results, NCCN recommended osimertinib for NSCLC patients with brain metastasis carrying EGFR sensitive mutations in the 3rd edition of guideline for Central Nervous System Cancers [29].

Multiple clinical trials indicate that benefits of osimertinib not only reflect in the management of T790M mutations. In 2020, FLAURA Phase III study comprehensively evaluates clinical benefits between osimertinib and first generation TKIs for untreated advanced-stage NSCLC with EGFR sensitive mutations. This study further confirmed the superiority of osimertinib in the management of all EGFR sensitive mutations. Compared to gefitinib or erlotinib, osimertinib could significantly extends PFS and reduces the incidence of CNS progression [30]. Importantly, a later published ADAURA study provided basis for the utilization of osimertinib as adjuvant therapy after surgical resection. Compared to placebo, osimertinib could bring significant benefits regarding DFS while the adverse event is within tolerance according to patients [31]. Besides, this strategy could also effectively reduce mortality and recurrence rate to nearly 20%. Even though the final OS has not been officially published, this result indicates a new era for NSCLC adjuvant therapy. However, the dosage, duration of treatment, and indications for this strategy should be further carefully studied before clinical application.

Furmonertinib (AST2818, alflutinib) is a third-generation EGFR-TKIs developed in China. Pre-clinical study indicates furmonertinib, as well as its metabolites in vivo, are highly selective anti-cancer agents [32]. Later, several Phase I clinical trials testified furmonertinib could effectively control the progress of advanced stage NSCLC with T790M mutations. Also, the adverse effects were within tolerance range according to patients’ report43. In 2021, a multi-centre, single-arm, phase IIb study included 220 cases of NSCLC patients who carriers primary T790M mutations or acquired T790M mutations after TKI treatment [33,34]. According to this study, the median PFS was 9.6 months, ORR was 74%, and tumor shrinkage was observed in 96% patients 44, 45. These results were slightly better, despite not statistically significant, than previous results in osimertinib study. This improvement could be attributed to the fact that both furmonertinib and its metabolites in vivo exhibited high anti-tumor activity, as is mentioned above. Meanwhile, similar to osimertinib, furmonertinib also has great permeability to Blood-Brain-Barrier. Clinical trial indicated the ORR for patients with cranial metastasis was 66% and mPFS was 11.6 months [34,35]. Therefore, furmonertinib is expected to be another treatment choice for T790M mutation, especially for patients with cranial metastasis, which is common among advanced stage NSCLC.

The efficacy of furmonertinib as first line therapy was also exhilarating. According to the recent published FURLONG study at 2022 European Lung Cancer Congress (ELCC), furmonertinib could significantly prolong PFS across all pre-specified subgroups compared to gefitinib as first line treatment in advanced-stage NSCLC patients. Meanwhile, the frequency of grade ≥ 3 treatment-related adverse events was 11% in furmonertinib and 18% in gefitinib group [36]. The final OS results of the FURLONG study was highly anticipated.

Aumolertinib (almonertinib, HS-10296) is another third-generation EGFR-TKIs developed in China. Preclinical studies confirm aumolertinib also has good blood brain penetration [37]. In the APOLLO Phase 1/2 trial, which includes patients with T790M mutation after EGFR-TKIs therapy, aumolertinib exhibited its efficacy based on ORR, mPFS, DoR, and OS [38,39]. Independent central review suggests This study also confirmed anti-cancer activity of aumolertinib in real patients. The CNS ORR was 60.9% and CNS DCR was 91.3% for patients with CNS metastases after taking aumolertinib [39]. Compared to gefitinib, first line aumolertinib treatment could also significantly prolong median PFS (mPFS) and DoR in Chinese patients with advanced EGFRm NSCLC according to the recent published AENEAS study. Moreover, aumolertinib has similar rate of adverse events that > grade 3 [40,41]. Currently, there are ongoing phase-III studies of almonertinib (NCT04687241) and furmonertinib (FORWARD, NCT03787992) aiming to evaluate efficacy of aumolertinib and furmonertinib as adjuvant therapy in EGFRm NSCLC patients after complete tumor resection [20]. We will learn more about the administration of these drugs in the future.

## 3. Drug Arrangement in Sequential Treatment-Maximizing Benefits for Patients

Based on these encouraging results, some scholars suggest that third generation EGFR-TKIs should be prescribed as first line treatment for patients carrying EGFR sensitive mutation regardless of T790M status. Indeed, this strategy was listed in the 2021 ASCO guideline and 2022 ESMO expert consensus for advanced or metastatic NSCLC [18,42,43]. Due to its good permeability across blood brain barrier, third-generation EGFR-TKIs should be prioritized for those patients with CNS, including leptomeningeal disease, as suggested by 2022 ESMO expert consensus [43]. However, on the one hand, from the perspective of the nations’ healthcare systems, using osimertinib as a first-line therapy is not cost effective and could bring significant burden to individuals and healthcare systems [44]. It should be noted that this conclusion was made based on the healthcare system in the US and Mexico, therefore, a detailed cost-effectiveness analysis of osimertinib as first line setting in China mainland and other regions is warranted.

Moreover, multiple retrospective real-world research indicated that utilization of osimertinib as a second-line therapy after first/second generation TKIs could delay the onset of resistance and effectively extend OS for patients, thereby maximizing the benefits to the patients [45,46]. In the LUX-Lung7 study, among the patients who received osimertinib after progression to afatinib, the mPFS was 21.9 m and 3-year OS rates were 90%. Importantly, for the patients receiving osimertinib as second-line treatment, the mPFS2 was even higher, reaching 53.3 months [47].

GioTag is the first clinical research aiming to evaluate appropriate drug regimen for patients carrying EGFR sensitive mutations worldwide. This real world, single-arm, multi-centre study retrospectively analyzes time on treatment (TOT) and OS for patients with EGFRm advanced or unresectable NSCLC who had T790M-positive disease after first-line afatinib and subsequently received osimertinib (2+3 treatment) [48]. The results indicate in afatinib plus osimertinib group, median time on treatment was 27.7 months while mOS was 37.6 months in whole population. In Asian subgroup, median time on treatment was 37.1 months while mOS was 44.8 months for afatinib plus osimertinib treatment. Importantly, the rate of cranial metastasis was stable prior and after afatinib treatment [48]. Therefore, fearing of potential cranial metastasis should not be considered as a reason for first line osimertinib treatment as osimertinib could effectively penetrate blood brain barrier. The feasibility of 2+3 treatment was further substantiated by the UpSwinG study published in December 2021. This observational study demonstrated promising activity of sequential afatinib and osimertinib in patients with EGFR-mutant NSCLC in 191 patients from 9 different countries [49]. Moreover, in the RESET study conducted in South Korean, the result indicates sequential afatinib and osimertinib treatment could improve survival rates, TOT, and DCR compared to treatment with afatinib followed by other chemotherapies. This study suggests such strategy could maximize the clinical benefits for patients while reducing chemotherapy exposure [50]. Given the real world setting of these studies, a head-to-head prospective trial that systematically compares mPFS, mOS, and TTF between sequential 2+3 vs osimertinib is urgently needed.

However, how to optimize the sequence of EGFR-TKIs treatment is also controversial. Whether using second generation followed by third generation EGFR-TKIs (2+3 strategy) or first generation followed by third generation EGFR-TKIs (1+3 strategy) is highly debated. Recent subgroup analysis of ARCHER 1050 study indicates even though dacomitinib could effectively extends OS compared to gefitinib in the whole population, the final mOS result was not statistically significant among Asians. Meanwhile, only 9.7% patients get the chance to receive osimertinib after resistance while this rate was 11.1% among gefitinib arms. Besides, the mOS for dacomitinib + osimertinib patients was also shorter than gefitinib + osimertinib patients [51,52]. However, a multi-centre retrospective study conducted in Japan suggests differently. The data indicates afatinib + osimertinib could brought significantly better ORR and disease control rate (DCR) compared to gefitinib + osimertinib. The mPFS, despite not statistically significant, also tends to be prolonged in afatinib + osimertinib arm [53]. Another small-scale retrospective research suggests for patients developed T790M mutation after administration of first/second generation EGFR-TKIs, receiving sequential afatinib + osimertinib could extends OS than receiving first generation EGFR-TKIs + osimertinib [54]. Some researchers attributed this improvement in efficacy to the fact that second generation EGFR-TKIs have a wider inhibitory spectrum compared to first generations, which means they could postpone clonal expansion and homogenize subclonal mutation, thereby delaying onset of acquired resistance and prolonging time on treatment. As a result, 2+3 strategy may be a better choice than 1+3 strategy.

Despite of the superiorities of second generation EGFR-TKIs mentioned above, it should be noted that compared to first generation EGFR-TKIs, adverse events are more severe and more often in second generation TKIs as they bind to EGFR in an irreversible manner [9]. Meanwhile, these results were limited by the sample size and nature of the retrospective analysis. The applicability and practicability of such regimen still need to be further explored in larger prospective research. Moreover, as Furmonertinib was recently approved by FDA, sequential treatment strategy for this drug lacks relevant support. Up till now, there was no clinical research nor retrospective study to systematically evaluate efficacy of afatinib plus furmonertinib or aumolertinib as first line treatment. Therefore, how to comprehensively evaluate mutational status and patients’ tolerance to adverse effects, optimize drug sequencing, and maximize the benefits of patients from third generation EGFR-TKIs will be a major field of study in the future. It should also be noted that in real world setting, only less than 30% of patients receiving first- or second-generation EGFR-TKIs as first line therapy have the chance to receive subsequent third generation EGFR-TKIs [52]. Therefore, comprehensively evaluating pros and cons of different first line therapy from different dimensions, including efficacy, improvement in QoL, adverse events, and cost-effectiveness, will be a major focus of research for our team in the future. Management of advanced stage lung cancer is a marathon rather than a sprint; hence oncologists nowadays need to take the long view rather than merely considering current situations. Establishing optimal management strategies for EGFRm NSCLC is still an unmet medical need that warrants further investigation.

## 4. Facing the Inevitable Rivalry: Mechanisms and Strategies against Them

### 4.1. EGFR Dependent Mutations

#### 4.1.1. C797S Mutations

Like T790M, C797S mutation also occurs at EGFR 20ex. It is reported that C797S mutation is the most common mechanism underlying on-target osimertinib resistance, developing in 10–26% of patients [1]. However, the exact mechanism of this phenomenon was not clearly elucidated. Crystallized structure indicates such mutation will only alter the hydrophilicity of 797 residue. Considering that C797 is the site of covalent binding for all known irreversible EGFR-TKIs, it is hypothesized such alteration will affect binding affinity of osimertinib and thereby resulting in re-activation of EGFR pathway [1,2,3]. In the absence of co-existing T790M mutations, which is common in patients receiving osimertinib as first line therapy, re-challenging with first- or second- generation EGFR-TKIs such as erlotinib and dacomitinib might be beneficial for the patients. However, the authors also suggested this strategy would eventually result in the co-existence of T790M and C797S [4].

Co-existence of T790M and C797S mutation is more common among patients receiving osimertinib as second-line therapy. Based on the allelic context of T790M and C797S, this mutation could be further divided into two subgroups, trans C797S and cis C797S mutation (Figure 2). An in vitro study published in 2015 indicated that for trans mutation, cells will be resistant to osimertinib but sensitive to a combination of first and third generation EGFR-TKIs. However, cells with trans mutations will be unresponsive to all EGFR-TKIs alone or in combination in vitro [5]. Therefore, different strategies were adopted to manage different types of C797S mutations in clinic. For cis mutations, there is a deficient of valuable meta-analysis or clinical research data to support clinical decisions. In 2018, Zhao et al. reported a case of an effectively managed lung cancer patient harboring triple EGFR mutations of L858R, T790M, and cis-C797S who was treated with a combination of osimertinib, bevacizumab, and brigatinib based on circulating-DNA mutational status [6]. In a small-scale retrospective study that included 15 cases of advanced NSCLC with cis C797S mutations, 5 patients received a combination of brigatinib (ALK inhibitor) and Cetuximab while 10 patients received platinum-based chemotherapy. The result suggests ORR and mPFS was 60% and 14 months in combinatorial group while only 10% and 3 months in chemotherapy group [3]. This research brought public’s attention on brigatinib-based therapy and multiple researchers published similar results [7,8]. However, based on our experience at daily clinical work at the National Cancer Center in China, the efficacy of this strategy is less than optimal, and it should be noted that these reports were neither large scale nor randomized trials. Therefore, the efficacy of brigatinib based therapy remains further exploration.

Trans C797S mutation only takes up a small proportion, approximately 10% among all C797S mutations [9]. Some reports suggested 1+3 combinatorial therapy, such as erlotinib plus osimertinib, could effectively reverse resistance [9,10,11]. However, limited to patient sample, this drug regimen was not widely accepted by many clinicians. Therefore, management strategy for such patients still lacks a unified standard. Further multi-centre clinical trials are needed to explore the efficacy and mechanism of such choice. The ORCHARD trial (NCT03944772) is a phase III multi-arm study aiming to evaluate the efficacy of osimertinib plus gefitinib in patients with C797S mutations after first-line osimertinib. We believe results from this trials will bring us insights regarding management of these patients in the future [12] (Figure 3).

Fourth generation EGFR-TKIs are developed aiming to overcome C797S mutations, which abolishes affinity between TKIs and EGFR. Even though clinical data are not available, many of them such as BPI-361175, TQB3804, and CH7233163 have shown their efficacy against C797S mutations in preclinical experiments and are undergoing phase I trials [13,14]. These data were highly anticipated by both clinicians and patients.

#### 4.1.2. EGFR Independent Mechanisms of Resistance

Histological transformation and bypass activation are the most frequent alterations in EGFR off target resistance to third generation EGFR-TKIs, including first- and second-line setting. We will discuss their mechanisms and corresponding clinical management.

#### 4.1.3. Small Cell Lung Cancer (SCLC) Transformation

In a small population of patients (5–10%) after EGFR-TKIs treatment, resistance is characterized by a transformation of histological type from NSCLC to SCLC [15]. However, the mechanisms underlying this transformation is undefined. One hypothesis suggests that SCLC originated from NSCLC, whereas some researchers argued this transformation is a state of intratumor heterogeneity and drug selection. In the other words, chemotherapy and TKIs effectively inhibit the growth of NSCLC, therefore resulting in SCLC dominance [16]. However, there was clear evidence that transformed SCLC is a unique type of SCLC, which is distinct from both SCLC and Combined SCLC [17]. Therefore, the first hypothesize was commonly accepted by most clinicians.

Immunohistochemistry indicators exhibited positive staining for CD56, Syn, TTF-1, and a strong positive staining for Ki-67, while a negative staining for PD-L1 after transformation [18]. Moreover, such transformation also indicates rapidly progressed diseases and a poorer prognosis, the OS after transformation is approximately 10 months [19]. Some researchers suggests ZEB1, SPP1, MUC1, CD44, and ESRP1 might involve in the transformation process based on GEO datasets, however, no in vitro or in vivo study was conducted to confirm this statement [20]. In 2020, E.Pros et al. suggests RB1 rearrangement in LUAD patients may be used to predict the risk of SCLC transformation under growth inhibition. Based on this conclusion, they proposed that RB1 status may be used as a marker for SCLC transformation [21].

In a recent retrospective analysis that includes 9 cases of patients with SCLC transformation in City of Hope, researchers suggested SCLC transformation patients have a unique histological, molecular, and clinical profile over various time points and therefore argues for a more precise classification based on the unique mutational status of each patients at different time points, making it more complicated to understand pathophysiological processes that leads to this transformation and develop valuable targets [15]. Currently, chemotherapy, especially the combination of etoposide and platinum is widely used for the treatment of transformed SCLC [19]. Combination of EGFR-TKIs and chemotherapy was also suggested by some clinicians as they can delay drug resistance and SCLC transformation, however, safety and efficacy of such combinatorial therapy is still doubtful. The utilization of immune checkpoint inhibitors (ICI) was also reported as case report in transformed SCLC patients, however, majority of researchers suggests the efficacy of ICI in transformed SCLC is far from ideal, this could be attributed to low level of PD-(L)1 expression in SCLC [17,22,23]. Apparently, more collaboration in research is needed to elucidate this process and formulate treatment strategies to overcome this issue.

#### 4.1.4. MET Amplification

MET is also a member of receptor of tyrosine kinase superfamily. Mainly expressed in mesenchymal and tumor cells, its overexpression could activate downstream RAS/RAF/MAPK pathway and lead to carcinogenesis [24]. The proportion of MET amplification within acquired resistance to EGFR-TKIs patients’ group is roughly 5–20%^59^. Preclinical in vivo study confirmed inhibition of MET pathway could re-sensitive resistant clones to gefitinib therapy [24,25]. Therefore, research in this area mainly focuses on the combinatorial therapy between EGFR-TKIs and MET inhibitors. And there are already results from phase I study to support such combinations. Crizotinib is a multitarget tyrosine kinase inhibitor that could binds to MET, ROS, and ALK. In preclinical and preliminary clinical trials, the combinatorial therapy of crizotinib and gefitinib could effectively reverse resistance caused by MET amplification [26,27]. A small-scale real-world study confirmed simultaneous inhibition of MET and EGFR could reverse EGFR resistance and extend PFS for patients. Generally, crizotinib was generally well-tolerated and effective in various studies across different populations [28]. Meanwhile, several selective MET inhibitors such as savolitinib and tepotinib also demonstrate their efficacy when combined with EGFR-TKIs. The INSIGHT study is a multicenter randomized trial aiming to evaluate the efficacy of tepotinib in patients with EGFR mutated NSCLC with MET amplification. The phase II result suggests for patients with high MET overexpression (IHC 3+), tepotinib + gefitinib could effectively extends mOS and mPFS, however, this improvement is not evident in patients with mild MET overexpression (IHC 2+) [29]. Even though this phase II study was prematurely terminated due to low recruitment, the efficacy of either tepotinib alone or in combination with other EGFR-TKIs in NSCLC patients with MET amplification still warrants further exploration. In a recent multicenter phase Ib study involving the efficacy of osimertinib + savolitinib on patients who harbor MET amplification and progressed after osimertinib treatment, interim data suggests such combination has satisfying risk-benefit profile for patients and anti-tumor activity [30]. However, these results are limited by the size of the study. Therefore, more detailed and comprehensive data needs to be published to confirm the plausibility of such strategy.

Amivantamab (JNJ-6372) is a bispecific EGFR and c-MET antibody that gains FDA approval for patients with locally advanced or metastatic EGFR exon 20 insertions NSCLC recently [31]. Recent study suggest it might be another therapy for patients progressed after osimertinib due to MET amplification. In the CHRYSALIS trial ORR was 36% among patients who progressed after osimertinib and 100% among TKI-naïve patients [31,32]. Recently, researchers from Korea, USA, and China initiated a phase III multi center trial aiming to compare the safety and efficacy of amivantamab + lazertinib versus osimertinib as first line treatment in EGFRm patients [33]. The result of this trial may dramatically change the paradigm of current management strategy for EGFRm NSCLC patients. Another phase III study of amivatamab with pemetrexed and cisplatin chemotherapy is also ongoing (NCT04538664). They may bring further surprise to our patients as well as us oncologists [34].

#### 4.1.5. HER2 Amplification

Osimertinib resistance induced by HER2 amplification was first reported by Planchard et al. in 2015 [35]. Interestingly, they discovered that for some reason, HER2 amplification and T790M are mutually exclusive [35]. osimertinib resistance due to HER2 amplification was approximately 3% in the FLAURA trials [36]. HER2 amplification is also a common mutation noticed in clinical trials of several other types of third generation EGFR-TKIs such as abivertinib [37]. Therefore, like MET, switching to other types of EGFR-TKIs could not effectively manage this situation. Considering HER2 amplification is a commonly spotted mutation a breast cancer and its antibody Trastuzumab could effectively inhibit breast cancer with such mutations. Therefore, the combinatorial therapy that includes Trastuzumab and EGFR-TKIs or Chemotherapy gains researchers’ attention. In 2018, a single arm phase II clinical trial that include 24 such patients indicated practicability of combining Trastuzumab and Paclitaxel. The ORR was 46% and response rate was highly correlated with HER2 expression level [38]. It should be noted that the control group was absent in this study, and the combination of Paclitaxel makes it impossible to determine the contribution of individual drugs regarding treatment efficacy. In 2021, Gan et al. reported several cases of successful management of HER2 induced osimertinib resistance using pyrotinib+EGFR-TKIs in a small-scale retrospective analysis [39]. However, this conclusion lacks evident data support considering sample type and design of the study.

#### 4.1.6. Overexpression of AXL

AXL is a member of TAM (TYRO3, AXL, and MER) family, which play important role in multiple cellular process such as growth, proliferation, apoptosis, and adhesion. Therefore, AXL overexpression is always an indicator of poor prognosis in tumor [40]. Activation of AXL Receptor and its ligand GAS6 is a major mechanism of acquired resistance to EGFR-TKIs therapy [41,42,43]. Both AXL and EGFR share the same PI3K/AKT and MAPK/ERK downstream pathway. Therefore, activation of AXL will bypass the inhibitory signal of EGFR-TKIs and continue to activate these pathways, leading to carcinogenesis. Meanwhile, overexpression of AXL could also promote Epithelium-Mesenchymal Transition (EMT), which would assist metastasis [44]. Unfortunately, there are no AXL inhibitors that was approved by FDA for NSCLC. Preclinical study suggests combination of AXL inhibitor and erlotinib could induce G2-M cell cycle arrest and enhance apoptosis for NSCLC that was resistant to EGFR-TKIs [41,44]. Even though the combinatorial utilization of AXL inhibitors and EGFR-TKIs exhibited its efficacy in vitro and in vivo, there are no clinical data to suggest the safety and efficacy of such regimen. Therefore, management of AXL-induced EGFR resistance is still a major challenge.

#### 4.1.7. Downstream BRAF^V600E^ Mutations

BRAF^V600E^ mutations mediated EGFR-TKIs resistance was first reported by Ho et al. in 2017, which was effectively controlled using BRAF and EGFR blockage simultaneously [45]. As is illustrated above, BRAF is located downstream of EGFR signaling cascade and it is estimated that BRAF^V600E^ occurs in 3% of patients who developed resistance to Osimertinib [46]. Up till now, little is known about this mechanism of resistance and no concensus has been agreed regarding management of these patients. In preclinical models, concurrent administration of BRAF inhibitor(encorafenib and dabrafenib) and osimertinib could effectively control resistance, meanwhile, the efficacy of this combination therapy was also substantiated by several case studies [9,45]. However, toxicity of this combination should not be overlooked since therapeutic agents targeting BRAF have been associated with numerous adverse events, serious adverse events which lead to dose reduction have been reported by several studies [47]. Therefore, the tolerability and potential benefit of this combination strategy should be carefully evaluated by physicians.

#### 4.1.8. Agnostic-Based Strategies: New Hopes

Recently, the concept of antibody-dependent drugs gained wide research attentions. Antibody-drug conjugates is a novel approach that might be effective in NSCLC patients developed resistance to TKIs [31]. Bound to a specific antibody, cytotoxic drug could thereby be delivered specifically into tumors and exerts its effect while minimizing unpleasant adverse events.

T-DM1 is an anti-HER2 antibody-drug conjugated trastuzumab emtansine. It is hypothesized that T-DM1 monotherapy could be effective in patients with concurrent EGFR and HER2 mutations. Preclinical study suggests combination of osimertinib and T-DM1 would effectively reverse off-target EGFR resistance caused by HER2 amplification [48]. Further research still needs to be done to evaluate effectiveness of these drugs. Overall, there are no approved therapy for EGFR mutation accompanied by HER2 amplification. A phase II clinical trial (TRAEMOS, NCT03784599) is now recruiting patients who harbors HER2 amplification and progressed after osimertinib treatment to evaluate possibility of osimertinib plus T-DM1 in clinical settings [49]. We believe we will get more information about how to manage these patients soon.

Patritumab Deruxtecan is an antibody-drug conjugate constituted by a HER3 antibody and topoisomerase I inhibitor. HER3 is one of the four members of EGFR family and considered as a compelling target to reverse resistance to EGFR-TKIs [50]. It is reported to be expressed in more than 83% of EGFRm lung cancers [51]. As molecular alterations leading to EGFR-TKIs resistance were diverse and targeting specific mutations to reverse resistance might be impractical in real world setting, therefore, targeting and delivering cytotoxic agents into tumor cells that expressed a ubiquitous receptor such as HER3 might provide a novel strategy for the management of TKI resistance for a broader population. A recent phase I study substantiated the safety and efficacy of Patritumab Deruxtecan for patients with EGFRm NSCLC that was resistant to previous EGFR-TKIs treatment. Among 57 patients, ORR was 39%, mDoR was 6.9 m, and mPFS was 8.2 m. The final OS data was still immature [52]. Previous similar studies suggested for patients receiving platinum-based salvage therapies after EGFR-TKIs resistance, mPFS was 2.8–3.2 m and mOS was only 7.5–10.6 m [53]. Even though adverse events of Patritumab Deruxtecan were observed in all participants and the proportion of ≥Grade 3 treatment related adverse events was 64%, the toxicity profile of Patritumab deruxtecan was still manageable, these serious toxicity effects could be successfully mitigated by dose reduction or dose delay in this clinical trial [52]. Considering this promising improvement in mPFS with acceptable adverse events, a global phase II trial (NCT04619004) aiming to evaluate safety and efficacy of Patritumab deruxtecan as a single agent for EGFRm NSCLC patients whose disease progressed after EGFR-TKIs treatment is ongoing [53].

Generally, antibody drug conjugate is a rapidly evolving area of study and appealing therapeutic options for a diverse subset of cancer. With a better understanding about the payload, linker, and the drug internalization process these years, antibody drug conjugate has become a “biological missile” target against cancer. Even though they exhibited promising effects in early phase clinical trials, several limits remain. Compared to conventional small molecules or chemotherapy, ADCs require a more complicated manufacturing process as well as a sophisticated drug component selection to maximizing their efficacy. Moreover, due to its relatively large molecular weight, the concentration of drug in tumors is limited. Study suggested only a small portion of ADCs could reach to the tumor cells. Therefore, efficacy of the efficient payload should also be considered when designing novel ADCs [54]. Overall, with a more elaborated molecular subtyping and pathological classification system of NSCLC, a better comprehension of the mechanisms of actions for ADCs, and a refinement of manufacturing protocols, we believe they will become powerful weapons for oncologists in the combat with cancer in the near future.

## 5. Future Directions

In the past decades, the development of EGFR-TKIs brought tremendous benefits to NSCLC patients as they could extend PFS and OS while having less adverse effects compared to traditional chemotherapy and radiotherapy. Their roles as neoadjuvant therapy also attracts wide attentions. However, obstacles still exist in this field, which limits its further utilization in clinical settings.

Different generations of EGFR-TKIs not only brought hope for patients, but also pose challenges to clinicians. How to maximize the efficacy of each type of EGFR-TKIs become a major issue in daily practices. It has been proved EGFR mutation is a highly heterogenous group, therefore, it is not sufficient to simply prescribe EGFR-TKIs to patients with EGFR mutations. The concept of precision oncology requires us to provide the best therapy to the suitable patients at the right time. Based on this idea, combinatorial therapy based on patient mutational status, biomarkers, and anticipated response to each generation of TKIs therapy is warranted. Our team as well as our collaborators worldwide have done tremendous work aiming to establish a comprehensive framework for molecular classification of NSCLC based on multi-omics data, however, there is still a long way to go.

Meanwhile, despite the emergence of osimertinib greatly controlled resistance due to T790M, C797S mutation was considered as a major mechanism of resistance for third-generation EGFR-TKIs. Even though some case reports provided experience about the utilization of brigatinib regarding management of different types of C797S mutations, a standardized drug regimen needs to be set to guide the clinical practices for these patients. The good news is various fourth-generation EGFR-TKIs dedicated to C797S mutations, such as EAI045 and TQB3804, were under development or undergoing clinical trials. However, an anticipated dilemma for clinicians would be what should we recommend to our patients harboring C797S mutation, brigatinib based therapy or fourth generation EGFR-TKIs clinical trial? Moreover, the ever evolving acquired resistance to TKIs also raised public’s concerns about super-resistant mutations. It would not be a surprise to us if we witnessed the emergence of another mutation that caused resistance to fourth generation EGFR-TKIs. Therefore, the priority of our treatment plan is still trying to delay the occurrence of drug resistance while bringing exact benefit to our patients.

Antibody drug conjugates represent an innovative strategy to overcome resistance and several clinical trials have demonstrated appealing results and additional research are ongoing. However, it should be reminded the dosage, schedule, indications, management of adverse events, drug choice after resistance, and lines of treatment should be carefully investigated before being widely accepted in clinics.

In conclusion, as NSCLC is a highly heterogenous cancer and resistance is an unavoidable demon that hinders survival of our patients, what we can do is to make the best use of our available drugs to delay the onset of resistance and overcome resistance, assisting our patients to get more benefits from treatment. With this in minds, how to plan drug choice for sequential therapy and adjust drug regimen based on patients’ phenotype as well as tumor microenvironment, managing to shape the evolution of NSCLC rather than attempting to chase the evolution of NSCLC would the future directions of NSCLC research as well as goals for clinicians.

## Figures and Tables

**Figure 1 biomedicines-11-01141-f001:**
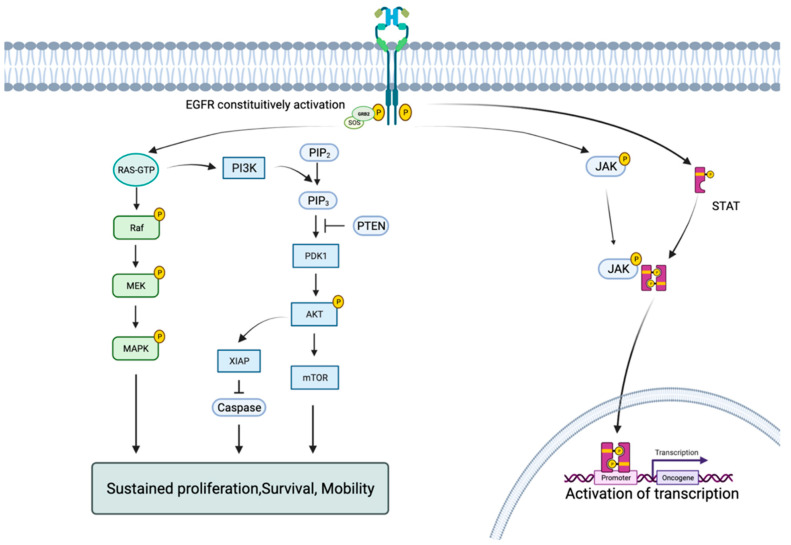
Summary of the EGFR-TKI signal transduction pathway.

**Figure 2 biomedicines-11-01141-f002:**
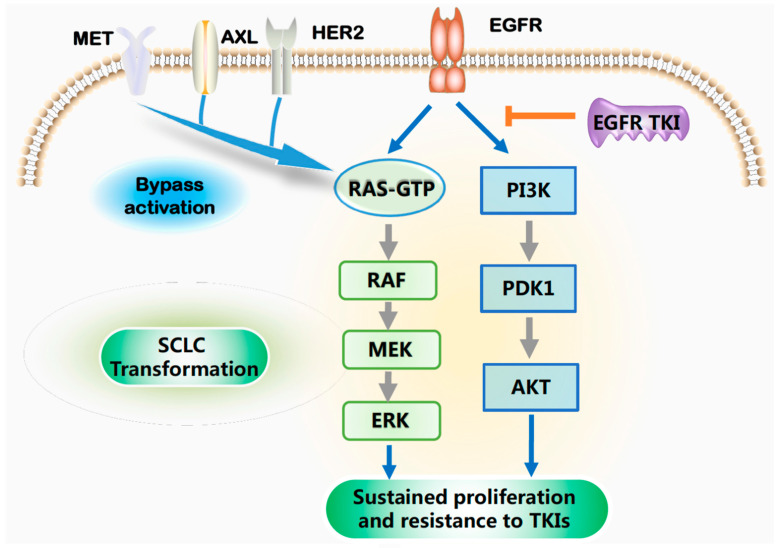
(**A**) Major resistance mechanism after EGFR-TKIs therapy. (**B**) Management strategies for different types of resistance.

**Figure 3 biomedicines-11-01141-f003:**
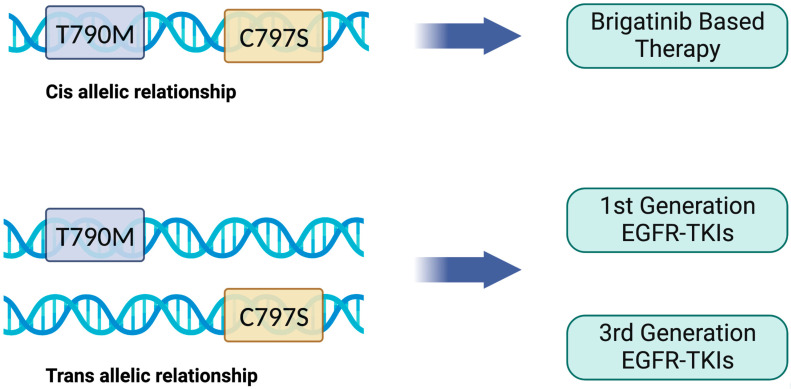
Illustration of the Cis C797S and Trans C797S mutation and their management.

**Table 1 biomedicines-11-01141-t001:** Overview of three generations of EGFR-TKIs and their indications.

Generic	Trade Name	Chemical Formula	Indications
Erlotinib	Tarceva^®^	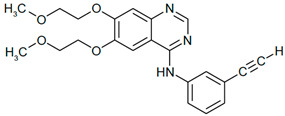	First-generation EGFR-TKIs, which could reversibly bind to EGFR. Sensitive mutations include 19del and L858R. They are now the first-line drug choice for EGFR-sensitive mutations. T790M mutations would likely arise and cause resistance.
Gefitinib	Iressa^®^	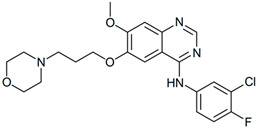	
Icotinib	Conmana^®^	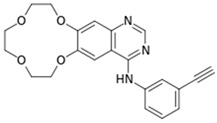	
Afatinib	Giotrif^®^	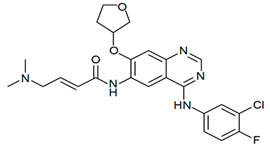	Second-generation EGFR-TKIs irreversibly bind to EGFR in a stronger manner. Effectively control major uncommon EGFR mutations. However, adverse events are more severe.
Dacomitinib	Vizimpro^®^	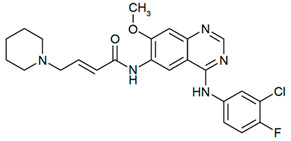	
Osimertinib	Tagrisso^®^	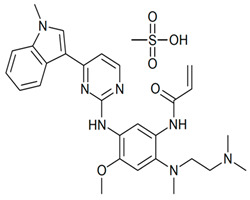	Third-generation EGFR-TKIs, which irreversibly bind to EGFR and effectively manage T790M mutation. Meanwhile, they can penetrate the Blood–Brain Barrier and thereby control CNS metastasis, providing more choice for the advanced stage.
Fumonertinib	Ivesa^®^	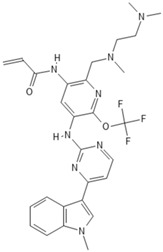	

## Data Availability

Not Applicable.

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
