# Peer review of "Making the Best Use of Available Weapons for the Inevitable Rivalry-Resistance to EGFR-TKIs"

_biomedicines, 2023, doi:10.3390/biomedicines11041141_

Round 1

Reviewer 1 Report

Lung cancer is the leading cause of cancer-related deaths worldwide, with non-small cell lung cancer (NSCLC) comprising 85% of lung cancers. T Targeted treatment of epidermal growth factor receptor (EGFR)-mutant non-small cell lung cancer (NSCLC) is a landmark for rational therapy addressing molecular vulnerabilities1. Treatment with first- and second-generation EGFR tyrosine kinase inhibitors (TKIs) markedly improved the clinical outcome of patients with advanced EGFR-mutant NSCLC. Currently, osimertinib is the only third-generation EGFR inhibitor approved for the sequential treatment of patients with acquired EGFRT790M resistance mutation occurring after first- and second-generation TKIs Despite the clinical efficacy of osimertinib in the first- and second-line treatment of EGFR-mutant NSCLC, drug resistance with disease progression is inevitable. Various EGFR-dependent and EGFR-independent resistance mechanisms have been identified includingn MET/HER2 amplification, activation of the RAS–mitogen-activated protein kinase (MAPK) or RAS–phosphatidylinositol 3-kinase (PI3K) pathways, new fusions, and histological transformation. RAS–MAPK pathway aberrations that are known to confer resistance to osimertinib include BRAF, NRAS, and KRAS mutations BRAF mutations occur in 2–4% of NSCLC patients and the vast majority are localized in the kinase domain, including the most common mutation BRAFV600E. BRAF mutations can be categorized into three classes based on their ability to act as monomers or dimers and based on their kinase activity. BRAFV600E mutations represent class I mutations that, similarly to class II BRAF mutations (RAS-independent), result in activation of the BRAF kinase and the MAPK pathway (gain of function). Class III BRAF mutations (RAS-dependent) result in an impaired BRAF kinase activity and amplify ERK signaling depending on upstream activating signals . All classes of BRAF mutations are recognized as oncogenic driver mutations, yet only BRAFV600 mutations represent clinically actionable drug targets in cancer patients.  BRAFV600E mutations have been identified as a resistance mechanism to osimertinib in roughly 3% of cases with EGFR-mutant lung cancer, with or without concurrent EGFRT790M mutation.  In this regard, it could be important to discuss  the clonal evolution of EGFR-mutant cells that concomitantly acquire BRAF mutations during anti-EGFR therapy.  This  review describes the research process of …the clinical evidence on the efficacy of third-generation EGFR-TKIs in patients with EGFR-mutated NSCLC. The theme of the article is very interesting and relevant. Basically, the paper has a lot potential. This paper has a clear structure.  The paper is pretty well written. The language is fluent and it is easy to read. The language follows an academic style. This is a quite good paper. I recommend publication after a minor revision

Author Response

Dear reviewer

Thank you for your patience and valuable suggestions

BRAFV600E did represent a major mechanism of resistance during EGFR-TKIs treatment, we apologize for our negligence that did not include this in our review. We have added a new section involving this mechanism and relevant management.

Best,

Reviewer 2 Report

A nice comprehensive review.  I would like to see a tabular presentation of these various agents with information on approved use in specific tumour types.

Author Response

Dear reviewer

Thank you for your patience and suggestions. 

Table1:An overview of three generation EGFR-TKIs and their indications is provided as a tabular presentation of these various agents with information on approved use in specific tumour types.

Best wishes!

Reviewer 3 Report

Thank you very much for the opportunity to review this interesting manuscript with an important clinical application. This is a well conduct review in the lung cancer treatment that discuss the sequential treatment aiming to delay the onset of resistance. I have a few commentaries that I believe can help improving the manuscript before accepting it.

Please review minor writing issues. 

- We generally don’t start a paragraph with “However” lines 198, 219. It is more used inside a paragraph to make a counterpoint.

- Follow the journal rules for spaces between lines

Include more key-words that are not acronyms to facilitate the manuscript finding. For example: “Lung Cancer”.

It will be important to include a flowchart to guide the treatment based on the success versus resistance – options to subsequent treatment.

Since this will be published in an International Journal Include some additional information about other populations, diagnosis and treatments that can be found in previous studies. Some suggestions: PMID: 35599008, PMID: 32575417, PMID: 33093800

Author Response

Dear reviewer:

Thanks for your kindness and patience providing valuable suggestions for our manuscripts. We have edited our manuscripts based on your suggestions.

1. Language and formatting issues have been revised.

2. More keywords are listed

3. Figure 2B and 3 are provided "as a flowchart to guide the treatment based on the success versus resistance – options to subsequent treatment".

4. additional information about other populations, diagnosis and treatments are listed in the revised manuscript.

Again, we sincerely thank you for your patience and your advice.